# Supply Chain Scheduling Optimization in an Agricultural Socialized Service Platform Based on the Coordination Degree

**Lingjingyuan Xu and Jianming Yao \***

School of Business, Renmin University of China, Beijing 100872, China
* Correspondence: jmyao@163.com

**Abstract:** In order to create a sustainable agricultural production system and meet the multi-stage and differentiated production needs of farmers, this study proposes to build an agricultural service platform to dispatch agricultural service providers. Reasonably handling the collaborative relationship between farmers and service providers is a key issue in platform scheduling. Based on the analysis of the operation characteristics of the agricultural service platform, this study redefines the core issue of handling the collaborative relationship—the coordination degree—from the special characteristics of agricultural services, then analyzes it in depth and proposes a portrayal method. On this basis, a multi-stage and multi-objective scheduling optimization model, which integrally reflects the service utility and service combination operation efficiency, is constructed, and an improved genetic algorithm is proposed for solving it. Then, this study designs a numerical experiment which describes the multi-stage decision making of farmers, and the simulation results show that the optimization model can provide a balanced multi-objective supplier scheduling solution for them. Further, the validity and feasibility of the model and algorithm are verified through comparative tests of optimization effects and sensitive analysis. This study contributes to research on sustainable agriculture by modeling collaboration between smallholder farmers and agricultural service providers, and provides an effective decision-making tool for agricultural service platforms.

**Keywords:** agricultural socialized service platform; coordination degree; supply chain scheduling; GA





## 1. Introduction

Since the reform and opening-up of China, the country's economy has continued to improve, and all sectors have made remarkable developments in terms of technological level and scale of production. However, compared with the development of China's overall economy during the same period, the development momentum of the primary industry is far less than that of the secondary and tertiary industries, and there are still many problems and obstacles in the process of modernization of agriculture. In order to better solve the current problems of unreasonable agricultural industry structure and poor supply channels of agricultural products, and to realize the sustainable development of the agricultural industry [1], the 19th CPC National Congress report proposed to "implement the strategy of revitalizing the countryside, improve the socialized agricultural service system, and realize the organic connection between smallholder farmers and modern agricultural development". In 2021, the Central Committee issued "No. 1 Document", which proposed "supporting various types of new agricultural service entities to carry out transformative large-scale services such as substitute farming, joint farming and land trust services" [2]. Therefore, the construction of agricultural socialized service system has been elevated to a new strategic level, which plays an important role in realizing the scale, systematization and organization of agricultural production.

Agricultural socialized services refer to agricultural production activities in which agricultural production subjects, mainly smallholder farmers, are able to obtain services from various agriculture-related economic organizations in society, thus overcoming the

disadvantages of their own small scale and obtaining the benefits of specialized division of labor and intensive service scale [3]. After years of support and guidance, various agricultural service organizations have flourished in the socialized agricultural service system, acting as service providers and providing multi-level and multi-stage agricultural services to agricultural production subjects. The problem of how to choose a more suitable service provider to provide services has become increasingly prominent, so the concept of the agricultural service platform has come into being. As an "integrated service provider", the agricultural service platform can effectively integrate services in all aspects of agricultural production and dispatch different service providers by providing service combinations, which can realize the specialization of division of labor, expand the scale of individual services to obtain scale benefits, and meet the diversified and differentiated production needs of farmers to achieve service [4]. By providing a combination of services, the agricultural service platform can achieve all-round coverage of farmers' production needs and consider synergistic effects among specific service operations to ensure effective connection between each stage of agricultural production activities. At the same time, the resource endowment of farmers in different situations (e.g., the scale of farmland, whether they are part-time workers, family labor situation, etc.) varies, and their recognition and acceptance of agricultural production services also differ. Agricultural service platforms should also take full account of the degree of collaboration between service providers and farmers when providing a portfolio of services, so as to maximize the benefits of local agricultural externalities and resources, and to change the small-scale, single-family production model of farmers that has predominated in the past, so that they can join the modern agricultural production model of specialized division of labor and intensive services.

Thus, this study intends to analyze the relationship between farmers and service providers in the process of receiving agricultural services, starting from the operational characteristics of the agricultural service platform, and build a multidimensional evaluation system of their collaboration degree accordingly. Based on this, the characteristics of agricultural production activities are analyzed in depth, a multi-objective multi-stage service supply chain optimization model is constructed, and the solution algorithm is discussed.

## 2. Characteristics of Agricultural Service Platform

The emergence of an agricultural service platform requires the development of an agricultural social service system to a certain extent, the existence of which depends on the richness of the service content of service providers on the one hand and the scale of farmers' needs on the other. At present, service providers providing agricultural productive services are diversified in terms of both subject types and service contents [5]. Many social capitals have entered the agricultural productive service industry and have certain markets and achievements [6]. For example, some enterprises have established technical service centers to provide agricultural production trusteeship services, and some enterprises have joined the agricultural products sales chain to open up a multi-channel sales layout. In addition, many agricultural machinery, agricultural materials, seeds and seedlings enterprises are competing to enter the agricultural production service market to meet farmers' diversified production needs [7]. In terms of service content, service providers can provide services in various agricultural production stages such as seed purchase, fertilization, pesticide application, machine plowing, processing and transportation, etc. In terms of service form, service providers can provide various element types of services such as technology, information, capital, agricultural insurance, etc. [8,9]. The service supply of multiple subjects, types, directions and stages provides the corresponding space of choice for the dispatch of agricultural service platforms.

At the same time, the service targets (i.e., farmers) faced by agricultural service platform services are heterogeneous among themselves. According to the differences in productivity, production methods and production purposes of agricultural producers, farm households can be divided into a precipitation layer consisting mainly of poor and weak farmers, a middle and lower layer consisting mainly of part-time farmers, an intermediate layer consisting of family farms and an upper layer consisting of large farms characterized by large-scale operations [10]. As there are differences in resource endowments among different types of farmers, the direction and characteristics of their demand for agricultural services are different; some may prefer service demand in traditional agricultural production processes such as breeding and fertilization, while others may prefer service demand in processes such as intelligent monitoring of farmland and farm machinery harvesting [11].

At the current level of agriculture in China, individual farmers with small-scale farming and decentralized farming operations are still the basic and main force in agricultural production [12,13], and although these individual farmers are willing to accept the provision of agricultural social services, there may be various reasons that prevent some of them from accepting such socialized services, such as not knowing which service providers are available or not being familiar with how to outsource a certain production, which prevents them from joining modern agricultural production [14,15]. Generally speaking, farmers may choose to join some agricultural cooperatives to access services, but there are some disadvantages to simply tying farmers to cooperatives. On the one hand, the services provided by individual agricultural cooperatives are limited and often confined to traditional agricultural production such as seeds and agricultural purchases, while value-added services such as deep processing of agricultural products and expansion of marketing channels are not available; on the other hand, agricultural cooperatives are usually linked by simple geographical attributes to form and provide uniform production services, which makes it difficult to meet the specific production needs of individual farmers [16].

The agricultural service platform can dispatch different service providers at different stages, provide corresponding services to farmers in the form of service combinations according to their agricultural production characteristics and their own resource endowments, and organically arrange and assemble the fragmented production tasks to obtain economies of scale. In this process, on the one hand, the agricultural service platform needs to select suitable service providers in each link of farmers' production operations, and consider whether the specific forms of services provided by them match farmers' needs and their own resources; on the other hand, the agricultural service platform also needs to consider the effective connection between different service stages, so that production tasks can be smoothly handed over between different service providers, so as to ensure the overall success of agricultural production.

In this way, smallholder farmers can effectively participate in the modernized and refined division-of-labor system of agriculture, change the inefficient production methods of the past, increase the output of agricultural products, improve the level of development of agricultural socialized services, drive the development of local agriculture, and truly realize the modern agricultural production model of specialized division of labor [17]. The operation mode of the platform is shown in Figure 1.

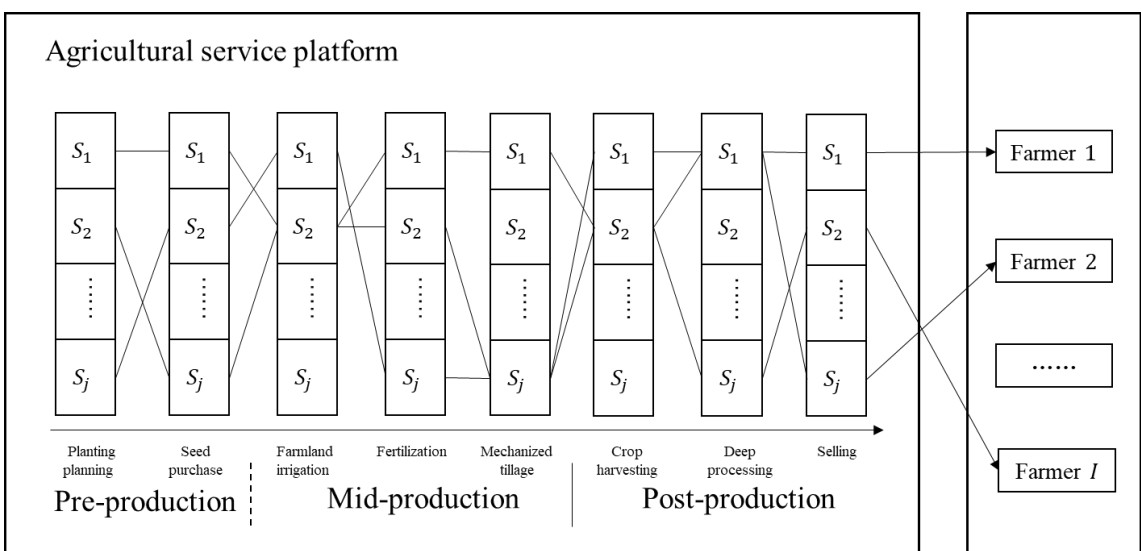

**Figure 1.** Service model of agricultural service platform.

## 3. Analysis of Coordination Degree between Agricultural Services and Agricultural Production

Unlike the role of the customer as a mere service recipient in conventional customized services, when a farmer outsources a production operation to a service provider and receives the corresponding service, the farmer is still not completely separated from the production process, which means that the final result of the operation is not entirely unilaterally dependent on the service quality of the supplier, but will be influenced by the degree of collaboration between the farmer and the service provider, which we call the coordination degree between agricultural services and agricultural production. In previous studies, coordination degree is usually used to express benign interaction in the coupling relationship of two macro systems [18]. Zhao (2018) pointed out that the coupling development of the modern service industry and modern agriculture is an objective requirement for adjusting and optimizing the industrial structure, and determined qualitatively that the coupling development of both is still at the primary stage [19]. Liu (2020) quantitatively analyzed the coupling and coordination degree of agricultural socialized services and agricultural modernization, and believed that the slight imbalance between them restricted the sustainable development level of the agricultural system [20]. Therefore, in an efficient and complete socialized agricultural service system, the coordination between service behavior and production behavior is critical. When the coordination between the two is high enough, it can give full play to the features of agricultural service intensification, specialization and scale, thus promoting the sustainable production of agricultural systems.

At the macro level, the coordination degree describes the relationship between the level of development of the integrated dimensions of agricultural services and agricultural production. When this system concept is translated into the micro level of a specific agricultural production operation, coordination degree is the extent to which the specific form of agricultural services fits with the farmer's own resources. For example, when a farmer has a demand for harvesting a certain crop, the specific harvesting service provided by a service provider may include large machinery harvesting, small machinery harvesting, manual harvesting, or loan outsourcing, etc. The farmers' acceptance of different forms of service varies due to their own resource conditions (e.g., cropland characteristics, family labor status, etc.). For example, farmers with irregularly shaped cropland have the best results in accepting small machinery harvesting or manual harvesting for harvesting operations. In the case of fertilizer application, the service provided by the supplier may be in the form of technical training, information on fertilizer ratios or direct assistance to farmers in fertilizer application, so farmers with part-time jobs may need fertilizer

application services more because of labor shortages, while farmers with training experience may only need information on specific fertilizer ratios to achieve better results.

Therefore, when farmers put forward certain production service demands at a certain stage, the platform should fully consider the degree of fit between farmers' own characteristics and the service forms provided by service providers. When the coordination degree between service providers and farmers is highest in the production activities by scheduling service providers, resource mismatch and waste can be effectively avoided. Using the professional and standardized socialized service power, farmers' own potential resources will be employed to improve their production efficiency and increase the output of the whole agricultural production system. Specifically, a higher coordination degree between farmers and service providers will reduce service costs on the one hand and improve the stability of the system on the other. Agricultural systems that incorporate the productivity of smallholder farmers will also achieve sustainable operation. In this case, smallholder farmers will be more efficient in tapping and utilizing resources, and therefore more connected to modern agriculture, thus achieving stable and sustainable production.

We assumed that in the agricultural production $m$ stage, there exists a service provider $j$ that $j \in \{1, 2, \dots J\}$, which can provide $p$ various specific types of services noted as $U_j = (u_{j1}, u_{j2}, \dots u_{jp})$. Since the service provider has certain business planning capabilities in its operation and is constrained by its own resource conditions, the specific ways it takes in fulfilling the service demand exist in the following probability distribution column vector $Prob_j = (p_{j1}, p_{j2}, \dots p_{jp})^T$, where $p_{jp} > 0$ and $p_{j1} + p_{j2} + \dots + p_{jp} = 1$. That is, when a service provider $j$ provides services to its client group, the specific service modalities available to farm households obey this probability distribution.

As mentioned earlier, there are differences in resource endowments among farmers, and these differences will lead to different levels of applicability of the specific service modality to the farmer, and therefore the efficiency of his collaboration with the service provider will vary when receiving production services from different service providers. Assuming that farmer $i$ can be characterized in terms of $q$, then the set of characteristics of farmer $i$ can be expressed as $V_i = (v_{i1}, v_{i2}, \dots v_{iq})$. In addition, the influence of each characteristic on the acceptability of farmers is different, so we set up the importance weights as $A_i = (a_{i1}, a_{i2}, \dots a_{iq})$, where $a_{iq} \geq 0$, and $a_{i1} + a_{i2} + \dots + a_{iq} = 1$. $a_{iq}$ denotes the weight of the $q$th characteristic dimension, reflecting the tradeoff of the degree of influence on each characteristic.

In turn, coordination degree matrix between farmer $i$ and service providers $j$ can be derived as $R_{imj(q \times p)} = \begin{pmatrix} r_{11} & \cdots & r_{1p} \\ \vdots & \ddots & \vdots \\ r_{q1} & \cdots & r_{qp} \end{pmatrix}$, the elements of the matrix (take $r_{qp}$ as an example) represent the applicability of farmer to the service approach $p$ in the $q$ dimension of the characteristic, which can be scored by an evaluation panel of experts on a scale of 1–5 from lowest to highest. Further, by considering the weights of the different dimensions of the farmer's characteristics, it is possible to obtain the degree of applicability of each service modality provided by the service provider $j$ $B_{imj} = A_i \times R_{imj} = (a_{i1}, a_{i2}, \dots$

$a_{iq}) \times \begin{pmatrix} r_{11} & \cdots & r_{1p} \\ \vdots & \ddots & \vdots \\ r_{q1} & \cdots & r_{qp} \end{pmatrix} = (b_{imj1}, b_{imj2}, \dots, b_{imjp})$. Taking into account the probability of each specific service modality provided by service provider $j$, the final score of coordination degree between farmer $i$ and service provider $j$ can be derived as $X_{imj} = B_{imj} \times Prob_j = (b_{imj1}, b_{imj2}, \dots, b_{imjp}) \times (p_{j1}, p_{j2}, \dots p_{jp})^T$.

## 4. Supply Chain Scheduling Analysis of Agricultural Production Services Portfolio Based on Coordination Degree

Due to the different production decisions of different farmers, their needs for agricultural production services vary greatly. In order to meet the differentiated needs of farmers'

production operations, the agricultural service platform can design a services portfolio containing different service items and dispatch service providers to achieve support for personalized and customized production operations for farmers at multiple levels and stages. Regarding service portfolio optimization, scholars generally agree that balanced optimization of multiple aspects of the service portfolio is needed. Chen (2010) proposed a service portfolio optimization method from the balanced demand of customers for service functions and quality [21]; Yao (2011) took fourth-party logistics as the research object and analyzed the revenue and risk decision of supply chain scheduling [22]; Liu (2020) considered high requirements of service products for dynamicity and proposed to optimize the articulation and cooperation of different suppliers in the service portfolio [23].

Based on the characteristics of agricultural production activities, the agricultural service platform, when scheduling service providers, needs to consider the service portfolio's utility and operation efficiency, in addition to the farmers' acceptance of the service form, i.e., the coordination degree between farmers and service providers as described in the previous section. In the following part, based on the coordination degree between farmers and service providers, we will analyze and characterize the utility and operation efficiency of the service portfolio, and establish the corresponding objective function to guide the scheduling decision of the service portfolio supply chain.

### 4.1. Analysis and Portrayal of Service Utility

In a study of farmers' demand for agricultural production services, Li (2015) pointed out that farmers prefer agricultural production services that can "save costs and increase efficiency" [24]. At the same time, the Ministry of Agriculture has been emphasizing the need to simultaneously promote quality and efficiency improvement and break the cost "floor" constraint in enhancing agricultural competitiveness and sustainable development. When farmers obtain the productive services they need, they usually judge the utility of the service in two ways: first, whether the effectiveness of the productive service meets their expectations, and second, whether the cost of obtaining the service is within their reach. For this reason, the following section analyses these two aspects and draws up the objective function.

(1) Effectiveness of service. The effectiveness of agricultural services is influenced by the timing of service delivery, in addition to the quality of services provided by the supplier. This is because agricultural production is a highly seasonal activity, in terms of sowing, seedling, harvesting, etc. These processes must be carried out within a certain time frame, otherwise it will greatly affect the subsequent activities or even bring negative effects such as yield reduction [25]. Therefore, when the platform schedules agricultural services, the key factor of whether services can be provided in a timely manner should be fully considered. Combined with the actual situation of agricultural production, farmers' requirements for the provision time of production services are not completely rigid. The fuzzy time window is a kind of time window describing customers' flexible service time preference [26], so this paper quantifies the impact of punctuality on service quality by fuzzifying the time of service provision by suppliers through the fuzzy time window affiliation function.

Assuming that the service provider is able to deliver the productive service on time during the corresponding farming season (i.e., the farmer's desired time) $[a_{imj}, b_{imj}]$, the punctuality of the service is 1, then the service quality will not be affected. When the service provider is not able to provide the productive service to assist the farmer to complete the production activity within the optimal farming season, but still within the acceptable time frame for the farmer, then the service quality will be compromised, but not to the extent that this production activity will be completely ineffective; otherwise, the service quality is 0. Thus, the quality of service in the agricultural productive service portfolio is influenced by the punctuality of service delivery. Using the fuzzy time window function, the punctuality of the service can be expressed as the fuzzy number of time $T$. Defining the service punctuality function as $\mu_{imj}$, the time of service delivery by the agricultural service

provider is $T_{imj}$, then the punctuality function of agricultural production services is shown in Equation (1).

$$
\mu_{imj}(t_i) = \begin{cases} 0, t_i < E_{imj} \\ (T_{imj} - E_{imj})/(a_{imj} - E_{imj}), E_{imj} < t_i < a_{imj} \\ 1, a_{imj} < t_i < b_{imj} \\ (T_{imj} - b_{imj})/(L_{imj} - b_{imj}), b_{imj} < t_i < L_{imj} \\ 0, t_i > L_{imj} \end{cases} \tag{1}
$$

Define the quality of the specific agricultural services provided by the service provider as $Q_{imj}$, considering the effect of punctuality on service quality, and the final service effect $F_{imj} = \mu_{imj}(t_i) \times Q_{imj}$.

(2) Cost of service. Another important factor to be considered in the provision of a production service portfolio to farmers is cost. As mentioned earlier, when a service provider provides a production service, the production activity is actually carried out in collaboration between the service provider and the farmer, and the higher the coordination degree between the two, the higher the acceptance of the production service by the farmer, thus mobilizing the farmer to make full use of their own resources and increase production efficiency, which ultimately leads to a reduction in the cost of the service. Based on the previous analytical portrayal of coordination degree, this section considers the quantification of the impact of coordination degree on costs. Since the coordination degree is a composite indicator that scores the degree of suitability between farmers and service providers, first we normalize it here, and the coordination degree is expressed as $X'_{imj}$, and $X'_{imj} \in [0, 1]$. Assume $\beta$ is the coefficient of the impact of the coordination degree on the cost, $\beta \in [0, 1]$. It is determined by a combination of factors such as the level of local agricultural production and the degree of completeness of the socialized service system. For example, in areas with a low level of agricultural production, resources are less fully utilized, so the higher the coordination degree, the more obvious it is for farmers to tap and mobilize their own resources, thus enabling farmers to significantly reduce the cost of services in collaboration with service providers, in this case, the value of $\beta$ should be appropriately adjusted upwards. Therefore, taking into account the coordination degree, the cost of a specific production service to the farmer should be $(1 - \beta X'_{imj})C_{imj}$.

### 4.2. Analysis and Portrayal of Service Efficiency

As the production needs of farmers are multi-stage and diverse, when providing service portfolios for them, an agricultural service platform should focus on the overall operational efficiency of the service portfolios, in addition to the specific utility of each service operation [27]. This ensures at the structural level that the agricultural production service system can continue to function effectively. When providing service portfolios for farmers, the agricultural service platform needs to effectively connect the services of each link to ensure the continuity of production activities, but also needs to reduce the internal and external uncertainties in the scheduling process and avoid the operational risks of the service platform [28]. Based on the above analysis, the agricultural production service platform needs to optimize its scheduling decisions in terms of both the fit and flexibility of the service portfolio when scheduling the service supply chain to provide the service portfolio.

(1) Service portfolio fit

Agricultural production activities are often multi-stage; the different stages interact with each other and are closely linked. Therefore, when providing a multi-stage agricultural service portfolio to farmers, this characteristic of agricultural production activities should be fully considered, and the degree of convergence between different stages (i.e., the degree of cooperation between service providers at different stages) should be improved to ensure that agricultural production activities are linked together in an orderly manner and run smoothly, and the degree of fit of the service portfolio should be pursued from the macro

perspective of the system as a whole. In turn, the service portfolio fit is determined by two aspects. On the one hand is the degree of cooperation between farmers and service providers, and on the other hand is the degree of cooperation between service providers at different stages.

The degree of collaboration between farmers and service providers is the coordination degree portrayed in the previous section, which will not be repeated here. This section will describe and portray the degree of collaboration between service providers. Due to the complexity of agricultural production, there is often no single supplier providing services at different stages, but agricultural activities are somewhat holistic in nature, and although the specific activities in the preceding and following stages are different, there is bound to be an undertaking relationship [29]. For example, if in the sowing stage, the service provider provides the corresponding service of turning the ground to start the gathering, then the specific means of operation of the subsequent irrigation and fertilization link form will have certain requirements and restrictions. Another example, in the storage phase after crop maturity, the specific storage methods provided by the service provider should also correspond to the subsequent marketing phase in order to enable the smooth implementation of the subsequent plan. It is foreseeable that even if a supplier at one stage is able to provide a more effective agricultural production service, if it is not sufficiently coordinated with service providers at other stages, this will lead to the fragmentation of agricultural production activities, which will naturally affect the final outcome of agricultural production.

In studies related to supplier collaboration, scholars have explored the factors that influence the stability of supplier collaboration, including trust, communication, and the degree of mutual involvement among others [30]. Based on these, we assume that the degree of collaboration between service providers $j$ and service provider $l$ is $\alpha_{jl}$, where $l \in \{1, 2, \ldots J\}$. Further, the degree of collaboration between the two is determined by the corporate relationship $e_1$, history of cooperation $e_2$, geographical distance $e_3$ and technical consistency $e_4$. Each specific factor can be judged through data collection and scored by a panel of experts, and to be consistent with the calculation of coordination degree, it is also expressed in five levels, scored from 1–5. The degree of coordination between service providers is then specifically calculated as $\alpha_{jl} = \omega_{e1}e_1 + \omega_{e2}e_2 + \omega_{e3}e_3 + \omega_{e4}e_4$. $\omega_{e1} \sim \omega_{e4}$ denote the relative weights of the four factors of corporate relationship, history of cooperation, geographical distance and technical consistency, respectively, which can be set by the agricultural service platform according to the specific circumstances of the service portfolio provided.

(2) Service portfolio flexibility

Agriculture has a longer life cycle than other industries, and there are more uncertainties in the environment which may lead to more risks. At the same time, the seasonal nature of agricultural production leads to a certain similarity of production rhythms among farmers, which may lead to a bunching of farmers' needs at a certain point in time, making it difficult for service platforms to schedule [31]. In addition, smallholder farmers, who are the main members in agricultural production, are fragile and less risk resistant, and can hardly withstand the adverse consequences of uncertainty [32]. To sum up, the agricultural service platform should consider enhancing the flexibility of the service portfolio when providing production services to farmers so as to reduce the impact of uncertainties in the environment and protect the interests of farmers and service providers. Therefore, when scheduling service providers, the flexibility of the offered service portfolio should be one of the optimization objectives, which will be analyzed and portrayed in this section.

The metric framework for supply chain flexibility is usually measured in terms of the ability to buffer, adapt, and innovate [33]. The higher the flexibility of the service portfolio, the more the service supply chain providing the service portfolio is able to respond to variable and complex multi-stage demand, and at the same time has a certain ability to resist the impact of external events. So, the flexibility of the service portfolio can be expressed in terms of resource surplus and risk resistance. Resource abundance $S_{imj}$

describes the ability of a service provider to meet the needs of diverse farmers, which is determined by the coordination degree between farmers and the service provider and the resource position of the service provider. The resource position of the service provider $V_j$ can be measured by three factors: the size of resources at its disposal $v_{j1}$; resource richness $v_{j2}$; and resource compatibility $v_{j3}$ [34]. It can be reflected by the size, richness and compatibility of the resources available. When the service provider's resources are sufficient and abundant, and the compatibility between different resources is high, the service provider can flexibly meet the diversified needs of different farmers, and when combined with the coordination between farmers and the service provider, the efficiency of resource utilization will be further enhanced, so the flexibility of the whole service portfolio will be increased. Then the resource abundance of the service provider in providing the corresponding services to the farmers $S_{imj}$ is specifically calculated as $S_{imj} = \gamma X'_{imj} \times V_j = \gamma X'_{imj} \times (\omega_{v1}v_{j1} + \omega_{v2}v_{j2} + \omega_{v3}v_{j3})$.

$\gamma$ is the influence coefficient of coordination degree on the efficiency of resource utilization, and $\omega_{v1} \sim \omega_{v3}$ are the relative weights of resource size, resource richness and resource compatibility, respectively, which can be set by the agricultural service platform according to the types and modes of services provided.

Sustainable operation of agricultural modernization systems requires good risk management, which is an important issue to consider when creating a socialized agricultural service system. Therefore, when dispatching service providers to provide services to farmers, an agricultural service platform should consider improving the flexibility of the service supply chain from the perspective of risk resilience, and use these social forces to reduce and share some of the agricultural risks [35]. For service providers $j$, the resilience of the service provider $G_j$ can be reflected in four aspects. The first is the technology level of the service provider $g_{j1}$. Advanced technology tends to reduce the probability of risk events. The second is the level of emergency management of the service provider $g_{j2}$. When a risk event occurs, the faster the service provider's response time and the more prepared the emergency supplies are, the more the losses from the risk event can be reduced. Thirdly, Government support is received by service providers $g_{j3}$. Government support, as a key player in agricultural risk management, can bring a great deal of security to agricultural production activities. Fourth is the level of insurance coverage of the service provider $g_{j4}$. This reflects the service provider's ability to use market mechanisms to resist risk [36]. Thus, the resilience of service providers $G_j$ is expressed as $G_j = \omega_{g1}g_{j1} + \omega_{g2}g_{j2} + \omega_{g3}g_{j3} + \omega_{g4}g_{j4}$. $\omega_{g1} \sim \omega_{g4}$ represent the relative weights of technical level, emergency management level, government support and insurance coverage, respectively, which can be set by the agricultural service platform according to the type and mode of services provided.

### 4.3. Scheduling Optimization Model

In this scheduling problem, we need to select the right service provider to serve the production needs of the farmer at a certain stage, so first we define the decision variables as $\theta_{imj}$. When supplier $j$ provides services to farmer $i$ at stage $m$, then $\theta_{imj} = 1$, otherwise $\theta_{imj} = 0$. We next normalize the parameters in the model by referring to the literature [37] to de-quantize them and use them in the multi-objective solution. The descriptions of the parameters and variables are detailed in Table 1.

**Table 1.** Description table of parameters and variables.

| Parameters and Variables | Description of Parameters and Variables |
|:---:|:---:|
| $i$ | Index of farmers |
| $m$ | Index of stages of agricultural production |
| $j, l$ | Index of Agricultural Service Providers |
| $\theta_{imj}$ | Decision-making variables, when supplier $j$ provides services to farmer $i$ at stage $m$, $\theta_{imj} = 1$, otherwise $\theta_{imj} = 0$. |
| $\mu_{imj}$ | Punctuality of service provider $j$ in providing services to farmer $i$ at stage $m$ |
| $Q_{imj}$ | Service Quality of service provider $j$ in providing services to farmer $i$ at stage $m$ |
| $X_{imj}$ | Coordination Degree of service provider $j$ in providing services to farmer $i$ at stage $m$ |
| $C_{imj}$ | Service Cost of service provider $j$ in providing services to farmer $i$ at stage $m$ |
| $\beta$ | Coefficient of impact of coordination degree on cost |
| $\alpha_{jl}$ | Degree of coordination between service provider $j$ and service provider $l$ |
| $S_{imj}$ | Resource Abundance of service provider $j$ in providing services to farmer $i$ at stage $m$ |
| $G_j$ | Resilience of service provider $j$ |

The optimization model is as follows.
Objective function:

$$\max Z_1 = \sum_{i=1}^{I} \sum_{j=1}^{J} \sum_{m=1}^{M} \mu_{imj}(t_i) \times Q_{imj} \times \theta_{imj} \tag{2}$$

$$\min Z_2 = \sum_{i=1}^{I} \sum_{j=1}^{J} \sum_{m=1}^{M} (1 - \beta X'_{imj}) \times C_{imj} \times \theta_{imj} \tag{3}$$

$$\max Z_3 = \sum_{i=1}^{I} \sum_{j=1}^{J} \sum_{m=1}^{M} X_{imj} \times \theta_{imj} + \sum_{i=1}^{I} \sum_{j=1}^{J} \sum_{l=1}^{J} \sum_{m=1}^{M-1} \alpha_{jl} \times \theta_{imj} \times \theta_{i(m+1)l} \tag{4}$$

$$\max Z_4 = \sum_{i=1}^{I} \sum_{j=1}^{J} \sum_{m=1}^{M} S_{imj} \times \theta_{imj} + \sum_{i=1}^{I} \sum_{j=1}^{J} \sum_{m=1}^{M} G_j \times \theta_{imj} \tag{5}$$

$$s.t. \sum_{j=1}^{J} \sum_{m=1}^{M} \mu_{imj}(t_i) \, Q_{imj} \times \theta_{imj} \geq Q_i^* \tag{6}$$

$$\sum_{j=1}^{J} \sum_{m=1}^{M} (1 - \beta X'_{imj}) C_{imj} \times \theta_{imj} \leq C_i^* \tag{7}$$

$$\sum_{i=1}^{I} \sum_{m=1}^{M} S_{imj} \times \theta_{imj} \leq S_j \tag{8}$$

$$\sum_{j=1}^{J} \theta_{imj} \leq 1 \tag{9}$$

There are four objective functions in the model. Equations (2)–(5) represent the four optimization objectives, which represent maximizing service effectiveness, minimizing service cost, maximizing service portfolio fit, and maximizing service portfolio flexibility, respectively; Equations (6)–(9) represent the four constraints, where Equations (6) and (7) represent that the total quality of the service portfolio received by farmer $i$ should be higher than his expectation and the total cost of the service portfolio should be lower than his expectation. Equation (8) represents the resource constraint of the service provider, i.e., the

demand it can satisfy cannot be higher than its resource reserve. Equation (9) represents the farmer $i$ can choose at most one service provider at each stage to obtain the corresponding production service.

## 5. Introduction to the Algorithm

The model in this paper is a typical multi-objective optimization problem with constraints, where it is not only the constraints that need to be considered, but also the tradeoffs between different objectives that may be in conflict. Such problems belong to NP-hard problems, and it often takes a long time to solve them using exact class algorithms when the problem size is large, so in solving such complex NP-hard problems, meta-heuristic algorithms are usually used, and stochastic algorithms are introduced on the basis of local search of heuristic algorithms to avoid falling into local optimum and search for global optimal solutions, such as forbidden search algorithms, simulated annealing algorithms, genetic algorithms, ant colony algorithms, etc. [28,37,38]. Since the decision variables of the model are 0–1 variables, for this kind of integer programming problem, genetic algorithms do not have the qualifications of function continuity and derivation, though they have better global search ability [39], while compared with other heuristic class algorithms, genetic algorithms are self-organizing, self-adaptive and self-learning [23], which are more suitable capabilities for the solution of the model in this study. Therefore, for the agricultural service supply chain scheduling problem based on coordination degree inscription, the genetic algorithm is selected to solve the problem in this paper, and the basic idea of GA design is referred to in the literature [40].

### 5.1. Treatment of Constraints

To enable the algorithm to achieve a satisfactory solution in effective time, the constraints (6)–(8) in the model are added to the objective function using the penalty function method in this paper, which is formulated as follows.

$$F_1 = \sigma \sum_{i=1}^{I} \left[ \max \left\{ 0, \ Q_i^* - \sum_{j=1}^{J} \sum_{m=1}^{M} \mu_{imj}(t_i) \, Q_{imj} \times \theta_{imj} \right\} \right]^2 \tag{10}$$

$$F_2 = \sigma \sum_{i=1}^{I} \left[ \min \left\{ 0, \ C_i^* - \sum_{j=1}^{J} \sum_{m=1}^{M} \left( 1 - \beta X'_{imj} \right) C_{imj} \times \theta_{imj} \right\} \right]^2 \tag{11}$$

$$F_3 = \sigma \sum_{j=1}^{J} \left[ \min \left\{ 0, \ S_j - \sum_{i=1}^{I} \sum_{m=1}^{M} S_{imj} \times \theta_{imj} \right\} \right]^2 \tag{12}$$

$$F = F_1 + F_2 + F_3 \tag{13}$$

$\sigma$ is a sufficiently large positive number, i.e., the penalty factor. When constraints (6)–(8) are all satisfied, the penalty function value is 0. Constraint (9) is a constraint on the value of the decision variable, which will be satisfied in the subsequent coding process.

### 5.2. Treatment of the Objective Function

In this paper, we use a linear weighting approach to transform the multi-objective optimization problem into a single-objective optimization problem. Since some of the four objective functions of the model require a maximum value and some require a minimum value, we first transform the objective function for the maximum value by taking the inverse to convert it into a minimum value problem. The objective functions (2), (4) and (5) are transformed into the minimum value problem of the following Equations (14)–(16).

$$\min Z'_1 = \frac{1}{\sum_{i=1}^{I} \sum_{j=1}^{J} \sum_{m=1}^{M} \mu_{imj}(t_i) \times Q_{imj} \times \theta_{imj}} \tag{14}$$

$$\min Z'_3 = \frac{1}{\sum_{i=1}^{I} \sum_{j=1}^{J} \sum_{m=1}^{M} X_{imj} \times \theta_{imj} + \sum_{i=1}^{I} \sum_{j=1}^{J} \sum_{l=1}^{J} \sum_{m=1}^{M-1} \alpha_{jl} \times \theta_{imj} \times \theta_{i(m+1)l}} \tag{15}$$

$$\min Z_4' = \frac{1}{\sum_{i=1}^{I} \sum_{j=1}^{J} \sum_{m=1}^{M} S_{imj} \times \theta_{imj} + \sum_{i=1}^{I} \sum_{j=1}^{J} \sum_{m=1}^{M} G_j \times \theta_{imj}} \tag{16}$$

Furthermore, considering the difference in magnitudes between different objective functions, the objective function is dimensionless in reference [40]. By assigning corresponding weights to each objective function, we can transform the multi-objective function into a single objective function. The weights among the objectives will change according to the different decision-making environments; for example, when farmers in a certain region generally grow crops with relatively mature and complete technologies, the service platform will focus on the cooperation of the service portfolio to further improve the efficiency of local agricultural operations on the basis of ensuring the utility of the service; then, when farmers in a certain region have a relatively weak resource base and poor risk resistance, the service platform will give priority to the service flexibility of the portfolio to guarantee the smooth production activities of the farmers. Therefore, the objective function of this model becomes the following form through linear weighting.

$$\min Z = a\frac{Z_1'}{minZ_1'} + b\frac{Z_2}{minZ_2} + c\frac{Z_3'}{minZ_3'} + d\frac{Z_4'}{minZ_4'} + F \tag{17}$$

$minZ_1'$, $minZ_2$, $minZ_3'$, and $minZ_4'$ are the minimum values obtained by the model when solving for a single objective, respectively. $a$, $b$, $c$ and $d$ are the weights of the four objective functions, respectively, and $a + b + c + d = 1$.

### 5.3. Genetic Algorithm Flow

Step 1: Initialize the population. In this study, we use a real number coding approach. The chromosome length depends on the total number of decisions of all farmers facing the dispatch, this is equal to the sum of the number of production stages to be decided for all farmers. When farmers have homogeneous production stages, this number is specifically $I \times M$. Take the value of the $im$ bit gene, for example—it signifies the index of service providers selected by the farmer $i$ at stage $m$.

Step 2: Genetic manipulation of parental populations, including crossover, mutation, generation of new populations, mixing to parental populations and calculation of fitness of each individual.

(1) Crossover: Crossover occurs by exchanging the information of two parents to provide a powerful exploration capability. We employ a one-cut-point crossover operator that randomly selects a cut point and exchanges the optimal part of the two parents to generate an offspring. In the evolutionary system, we first set the crossover probability to the parameter $P_c$. Because the optional suppliers are not necessarily the same for different farmers at different stages, we adopt relative sequential numbering for supplier numbering and set up an adjustment procedure to ensure that the decisions of the exchanged gene representatives are always in the optional range.

(2) Mutation: Modify one or more gene values from an existing individual to cause mutation. The mutation operator's function is to randomly select some parents and alter some of their genes with a probability equal to the mutation rate. In the evolutionary system, we first set the mutation probability to the parameter $P_m$ and decrease the number of mutated genes with each iteration.

(3) Evaluation: In nature, it is necessary to provide a driving mechanism for better individuals to survive. Our assessment consists of associating each chromosome with a fitness value to demonstrate its value on the basis of achieving the objective function. The higher the fitness value of an individual, the greater its chances of survival in the next generation. Therefore, individuals are selected as parents of the next generation based on their fitness values. After obtaining all fitness values of the chromosomes, a roulette-wheel approach is used in the selection process.

Step 3: Roulette and elite strategies are used to select chromosomes and retain the best chromosomes in each generation to form the next generation population.

Step 4: Determine whether to stop. When the number of iterations of the program reaches the set maximum number of iterations, the operation is stopped.

The programming logic of the genetic algorithm is shown in Figure 2.

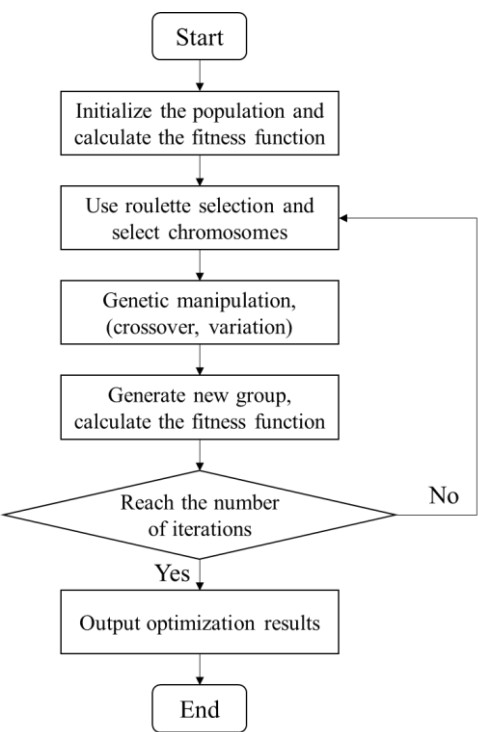

**Figure 2.** Genetic algorithm programming logic.

## 6. Analysis of Algorithms

### 6.1. Numerical Experiments

In this study, we assume that the farmers' decisions are independent of each other, so the following simulation will take the full-stage agricultural production activities of one farmer as an example. The crop he grows requires the purchase of corresponding services at six stages: seed purchase, sowing and seedling, farmland irrigation, fertilizer application and spreading, crop harvesting, and product selling, and we use $m_1 \sim m_6$ respectively. There are eight agricultural service providers in this farmer's county, which are denoted by $j_1 \sim j_8$. Among these service providers, $j_1 \sim j_4$ are integrated agricultural service providers which can provide the corresponding agricultural services to the farmer at all stages, while $j_5 \sim j_8$ are the specialized category of agricultural service providers, which can provide the corresponding services for some stages of agricultural production. The specific range of services provided by each agricultural service provider is shown in Table 2.

**Table 2.** Scope of services provided by service providers.

|       | $j_1$ | $j_2$ | $j_3$ | $j_4$ | $j_5$ | $j_6$ | $j_7$ | $j_8$ |
|-------|-------|-------|-------|-------|-------|-------|-------|-------|
| $m_1$ | √ | √ | √ | √ | √ |   | √ |   |
| $m_2$ | √ | √ | √ | √ | √ |   | √ |   |
| $m_3$ | √ | √ | √ | √ | √ | √ |   |   |
| $m_4$ | √ | √ | √ | √ | √ | √ |   |   |
| $m_5$ | √ | √ | √ | √ |   | √ |   | √ |
| $m_6$ | √ | √ | √ | √ |   | √ |   | √ |

The scheduling program operational parameters are calculated based on the multidimensional evaluation indicator system described earlier, with Table 3 showing the

corresponding parameters for each phase of the agricultural service provider's service delivery and Table 4 showing the service provider's operational parameters, including the degree of fit with other service providers and risk resilience. All parameters have been de-unitized and normalized.

**Table 3.** Corresponding parameters of service providers' provision of services.

| Stage | Service Provider | Service Effectiveness | Service Cost | Coordination Degree | Resource Abundance |
|-------|------------------|-----------------------|--------------|---------------------|--------------------|
| $m_1$ | $j_1$ | 0.46 | 0.7 | 0.69 | 0.45 |
| | $j_2$ | 0.49 | 0.61 | 0.64 | 0.55 |
| | $j_3$ | 0.53 | 0.64 | 0.61 | 0.54 |
| | $j_4$ | 0.59 | 0.61 | 0.68 | 0.49 |
| | $j_5$ | 0.64 | 0.76 | 0.74 | 0.65 |
| | $j_7$ | 0.87 | 0.82 | 0.32 | 0.73 |
| $m_2$ | $j_1$ | 0.59 | 0.7 | 0.64 | 0.41 |
| | $j_2$ | 0.47 | 0.6 | 0.62 | 0.54 |
| | $j_3$ | 0.6 | 0.68 | 0.64 | 0.46 |
| | $j_4$ | 0.59 | 0.68 | 0.61 | 0.41 |
| | $j_5$ | 0.73 | 0.74 | 0.81 | 0.67 |
| | $j_7$ | 0.89 | 0.84 | 0.58 | 0.77 |
| $m_3$ | $j_1$ | 0.52 | 0.69 | 0.61 | 0.58 |
| | $j_2$ | 0.57 | 0.61 | 0.69 | 0.56 |
| | $j_3$ | 0.47 | 0.64 | 0.7 | 0.47 |
| | $j_4$ | 0.51 | 0.63 | 0.66 | 0.49 |
| | $j_5$ | 0.72 | 0.75 | 0.73 | 0.65 |
| | $j_6$ | 0.64 | 0.74 | 0.8 | 0.63 |
| $m_4$ | $j_1$ | 0.59 | 0.68 | 0.61 | 0.48 |
| | $j_2$ | 0.57 | 0.64 | 0.62 | 0.36 |
| | $j_3$ | 0.59 | 0.69 | 0.64 | 0.39 |
| | $j_4$ | 0.55 | 0.62 | 0.68 | 0.44 |
| | $j_5$ | 0.69 | 0.71 | 0.73 | 0.66 |
| | $j_6$ | 0.72 | 0.72 | 0.76 | 0.62 |
| $m_5$ | $j_1$ | 0.49 | 0.63 | 0.6 | 0.37 |
| | $j_2$ | 0.53 | 0.61 | 0.6 | 0.55 |
| | $j_3$ | 0.55 | 0.61 | 0.62 | 0.36 |
| | $j_4$ | 0.58 | 0.69 | 0.66 | 0.37 |
| | $j_6$ | 0.71 | 0.71 | 0.79 | 0.65 |
| | $j_8$ | 0.82 | 0.8 | 0.53 | 0.76 |
| $m_6$ | $j_1$ | 0.59 | 0.66 | 0.67 | 0.35 |
| | $j_2$ | 0.53 | 0.65 | 0.66 | 0.37 |
| | $j_3$ | 0.47 | 0.61 | 0.65 | 0.43 |
| | $j_4$ | 0.47 | 0.69 | 0.65 | 0.39 |
| | $j_6$ | 0.66 | 0.72 | 0.82 | 0.65 |
| | $j_8$ | 0.79 | 0.89 | 0.45 | 0.8 |

**Table 4.** The degree of cooperation between service providers and risk resilience parameters.

| | Risk-Resilience | $j_1$ | $j_2$ | $j_3$ | $j_4$ | $j_5$ | $j_6$ | $j_7$ | $j_8$ |
|-------|-----------------|-------|-------|-------|-------|-------|-------|-------|-------|
| $j_1$ | 0.86 | 1 | 0.99 | 0.99 | 0.91 | 0.84 | 0.86 | 0.68 | 0.7 |
| $j_2$ | 0.82 | 0.99 | 1 | 0.98 | 0.93 | 0.72 | 0.84 | 0.72 | 0.74 |
| $j_3$ | 0.82 | 0.99 | 0.98 | 1 | 0.93 | 0.83 | 0.88 | 0.67 | 0.71 |
| $j_4$ | 0.89 | 0.91 | 0.93 | 0.93 | 1 | 0.8 | 0.88 | 0.65 | 0.71 |
| $j_5$ | 0.62 | 0.84 | 0.72 | 0.83 | 0.8 | 1 | 0.97 | 0.72 | 0.74 |
| $j_6$ | 0.7 | 0.86 | 0.84 | 0.88 | 0.88 | 0.97 | 1 | 0.7 | 0.73 |
| $j_7$ | 0.77 | 0.68 | 0.72 | 0.67 | 0.65 | 0.72 | 0.7 | 1 | 0.58 |
| $j_8$ | 0.81 | 0.7 | 0.74 | 0.71 | 0.71 | 0.74 | 0.73 | 0.58 | 1 |

### 6.2. Results of the Algorithm

The agricultural social service supply chain scheduling optimization problem portrayed in this study is a multi-objective optimization problem, including maximization of service effectiveness, minimization of service cost, maximization of service portfolio fit and maximization of service portfolio flexibility. Generally speaking, the setting of multi-objective weights for supply chain scheduling requires the service platform to make a comprehensive determination from the positioning and objectives of scheduling, based on the full consideration of the long term and the near term, local and overall balance, and the specific problems to be solved by scheduling. In the case of this paper, based on the resources of suppliers that the service platform can obtain, the feedback of farmers' demands for production services and the service platform's acceptability of the coordination process, the agricultural service platform puts the focus of scheduling on the optimization of service effects to effectively achieve cost savings and efficiency gains in farmers' production in the current year, out of the consideration of the conservative estimation of risks in the current year and the strategic goal of pursuing near-term results, while the other two objectives are relatively weakened, so the multi-objective weights are set to [0.3; 0.3; 0.2; 0.2]. All computations are implemented in MATLAB_R2020a on a Win64 laptop with core(TM) i7-8656U CUP @ 2.40 GHZ Intel processor with 16 GB RAM. After several iterations, the more suitable algorithm parameters were determined to be: population size 300, crossover probability 0.9, mutation probability 0.1, and number of iterations 200, which are shown in Table 5.

**Table 5.** Parameters of GA.

| | Pop Size | Maximum Generations | Crossover Probability $P_c$ | Mutation Probability $P_m$ |
|---|---|---|---|---|
| Value | 300 | 200 | 0.9 | 0.1 |

The average computation time is 25.22 s for 50 calculations of the arithmetic case, with high computational efficiency; when convergence is reached, the average value of the optimal fitness is 1.138844, the maximum value is 1.152477, and the minimum value is 1.131648, with deviations from the average value of 1.20% and 0.63%, respectively, with relatively stable computational results and good robustness of the algorithm. In this example, the optimal scheduling scheme corresponds to four objective function values of 4.37, 4.51, 8.19 and 8.44 for service effectiveness, service cost, service portfolio fit and service portfolio flexibility, respectively, Figure 3 shows the convergence of model adaptation, Figure 4 shows the convergence of the four objective functions, and Table 6 shows the optimal scheduling scheme.

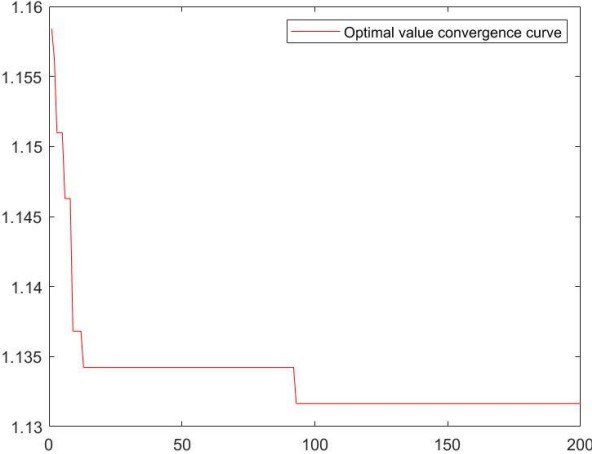

**Figure 3.** Model adaptation convergence results.

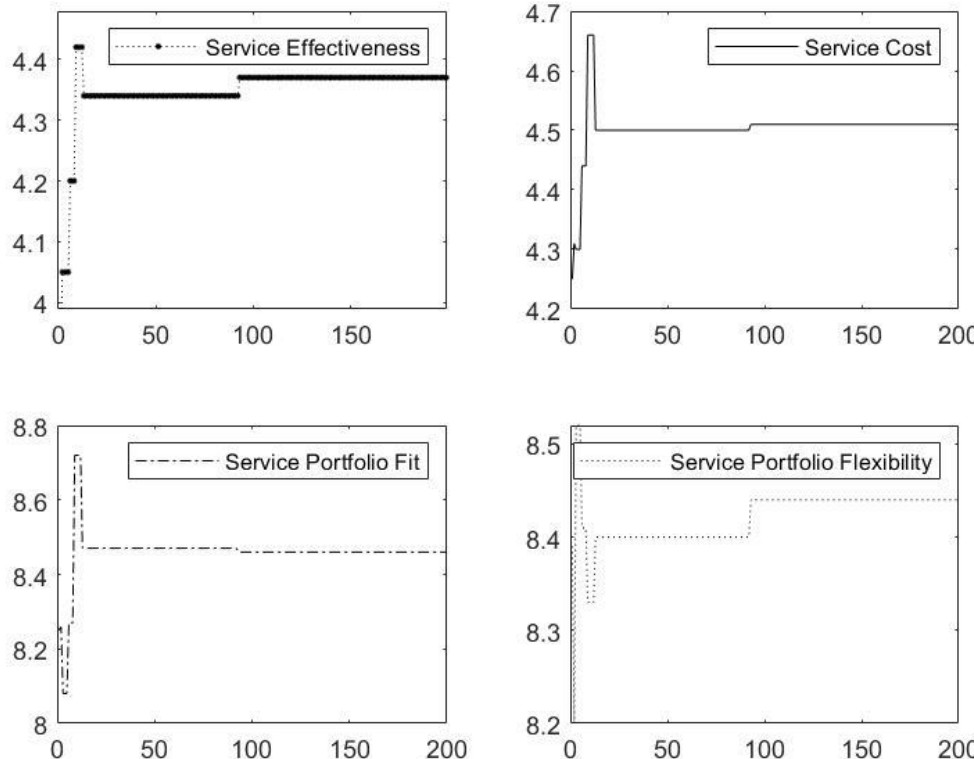

**Figure 4.** Convergence result of the objective function.

**Table 6.** Optimal scheduling options.

|  | $m_1$ | $m_2$ | $m_3$ | $m_4$ | $m_5$ | $m_6$ |
|---|---|---|---|---|---|---|
| Supplier Selection | $j_4$ | $j_5$ | $j_5$ | $j_6$ | $j_8$ | $j_8$ |

The scheduling scheme gives the optimal agricultural service provider for each stage of agricultural production, with service provider $j_4$ providing the first stage of seed purchase, service provider $j_5$ providing the second stage of seed planting and irrigation, service provider $j_6$ providing the fourth stage of fertilizer application, and service provider $j_8$ providing the fifth stage of crop harvesting and the sixth stage of product sales.

Without an agricultural service platform that integrates the resources of suppliers to provide one-stop all-stage services to farmers, farmers may only receive services through a single service provider because they have less access to information or difficulty in making judgments. Through the scheduling of the agricultural service platform, the farmer can receive more appropriate special services at different stages and obtain the best overall results. Assuming that the farmer can initially choose only $j_1$, $j_2$, $j_3$ or $j_4$ integrated agricultural service providers to obtain services, we compare the objective values of the optimal solution and the single provider selection solution. We can see in Figure 5 that the optimal scheduling scheme is significantly better than the initial scheme, with an average reduction of 34.49% in the objective value. Thus, the method is able to bring an increase in production for farmers.

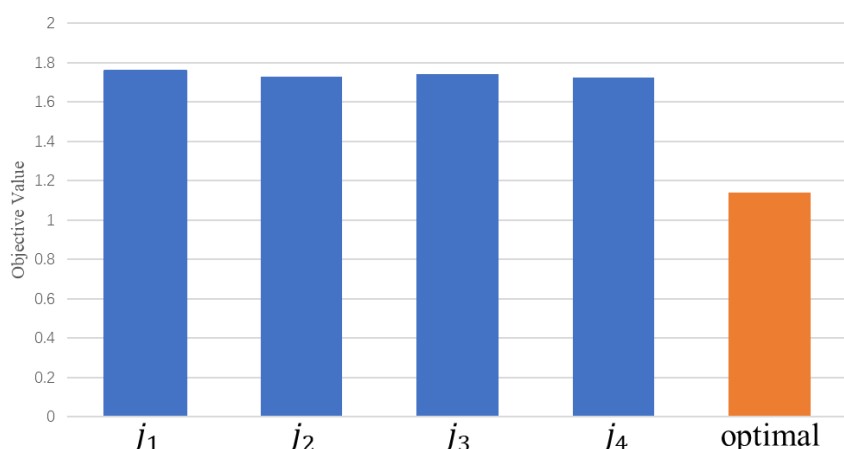

**Figure 5.** Comparison of optimal and original scheduling scheme.

Generally speaking, integrated service providers provide more moderate service effectiveness, but they are able to ensure a smooth connection between different stages of agricultural activities due to their wide scope of business, while specialized service providers only provide services in some stages of agricultural production, which are more effective but difficult to coordinate with suppliers in the upper and lower stages. The optimal scheduling scheme includes both integrated and specialized service providers, thus balancing service effectiveness and efficiency of the service portfolio to a certain extent. Service provider $j_5$, $j_6$ and other service providers are better coordinated, so in the medium-term agricultural production activities, both of them are preferred to ensure the smooth operation of the service portfolio, which shows the rationality of this scheduling scheme. In general, the model and algorithm in this study give better solution results for the multi-stage agricultural social service supply chain scheduling problem based on the coordination degree.

### 6.3. Sensitive Analysis

For different regions or different types of agricultural activities, the service platform faces different production demands and resource endowments, so the importance of service effectiveness, service cost, service portfolio fit, and service portfolio flexibility need to be adjusted accordingly. The service platform should set dynamic weights for the four objectives according to the specific situation when conducting supply chain scheduling. Here we compare the optimal scheduling options under five different weight allocation scenarios [0.25; 0.25; 0.25; 0.25], [0.7; 0.1; 0.1; 0.1], [0.1; 0.7; 0.1; 0.1], [0.1; 0.1; 0.7; 0.1], [0.1; 0.1; 0.1; 0.7] as shown in Figure 6. In scenario 1 we assign the same weight to the four objectives, while in scenarios 2–5 we assign a larger weight to Z1, Z2, Z3, Z4, respectively. From the figure we can see that when the four objectives are given equal weight, their optimization values are in the middle level of the five scenarios. When other conditions remain unchanged, increasing the weight of one objective function can significantly change the optimization value. Since Z1, Z3, Z4 are maximizing optimization objectives, the triangles on the Z1, Z3 and Z4 lines indicate that the maximum value of the corresponding objective appears in the scenario that gives it a larger weight of 0.7. Similarly, Z2 as the minimizing optimization objective, the triangle on the Z2 line indicates that the minimum value of Z2 among the five scenarios appears in scenario 3, with a weight of 0.7 to Z2. The above demonstrates the impact of weight on the objective's value. Therefore, the service platform can achieve a satisfactory scheduling option by adjusting the weights of the objectives according to the specific situation.

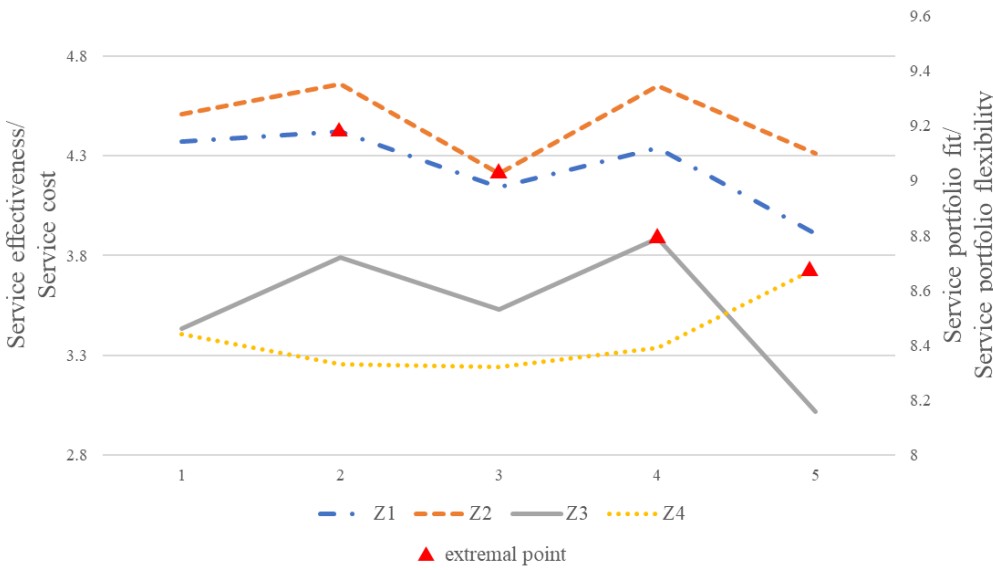

**Figure 6.** Change of the objectives under different weights.

## 7. Conclusions

The construction of an agricultural social service system has received increasing attention and concern in the process of agricultural modernization, and the participation of a wide range of social agents in the production process of agriculture to provide services to agricultural producers is a prerequisite and basis for enhancing agricultural intensification and specialization. However, agricultural producers simply obtaining social services in one or some stages cannot form an effective system, nor will they form a truly modern agricultural production structure, and the effect of production efficiency enhancement is limited. Therefore, this study considers the problem of agricultural service supply chain scheduling from the perspective of the overall system, relying on agricultural service platforms, and provides innovative organizational form ideas for the scale-up of agricultural productive services.

Firstly, starting from the special characteristics of providing agricultural services, this study points out the collaborative relationship between farmers and service providers, thus emphasizing the importance of the degree of matching the two in resources, i.e., coordination degree, and proposes a multidimensional evaluation system of coordination. Secondly, based on the inscription of the coordination degree, this study combines the characteristics of agricultural production activities and constructs a multi-objective and multi-stage service supply chain optimization model. Thirdly, the genetic algorithm is designed and improved according to the model, and the effectiveness of the model and algorithm is better verified.

With the gradual enrichment and diversification of subjects in the agricultural socialized service system, the effective dispatch and allocation of resources to the most appropriate places is the key to further improving the operational efficiency of the agricultural socialized service system. In future research, we will further consider the dynamic interaction of subjects in the agricultural socialized service system and conduct in-depth analysis and research on the optimization of service supply chain scheduling in the special context of agricultural production.

**Author Contributions:** Writing—original draft, L.X.; Writing—review & editing, J.Y. All authors have read and agreed to the published version of the manuscript.

**Funding:** This work was supported by the National Natural Science Foundation of China [grant number 71872174].

**Institutional Review Board Statement:** Not applicable.

**Informed Consent Statement:** Not applicable.

**Data Availability Statement:** Not applicable.

**Conflicts of Interest:** The authors declare no conflict of interest.

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
