# Peer review of "Supply Chain Scheduling Optimization in an Agricultural Socialized Service Platform Based on the Coordination Degree"

_sustainability, doi:10.3390/su142316290_

Round 1

Reviewer 1 Report

Thanks to the authors' innovative efforts in investigating the role of agricultural socializing platforms in reallocating resources to improve system efficiency by considering collaborative relationships between farmers and service providers and characteristics of agricultural production activities.

Reviewer 2 Report

1.     This paper presents a multi-stage and multi-objective scheduling optimization model for a sustainable agricultural production system. A case study of six stages and eight service providers is used to find the optimal agricultural service provider for each stage of agricultural production. This is a well written paper. But some minor points need to be improved.

2.     In section 4, all equations in the four objective functions need to be explained or cited a reference. How to determine these four objective functions including maximizations of service effectiveness, minimization of service cost, maximization of service portfolio fit and maximization of service portfolio flexibility? Is any objective function related to sustainability issue?

3.     Please cited GA algorithm from which reference./ The model parameters obtained using GA in Section 6 should be provided.

4.     Typo error “Table 6 shows the optimal scheduling scheme.” No Table 6? It should be Table 5.

5.     The optimal scheduling scheme should compare the original scheduling scheme. 

     6.  In this numerical experiment, the proposed method should compare with some existing methods or provide a sensitivity analysis. 

     7. The weights in Equation (17) should be addressed.

     8. All references need to be checked. For example:

"26. Wang M, Y. J. "Intertwined supply network design under facility and transportation disruption from the viability perspective." International  Journal of Production Research 4 (2021): 1-31." is not correct.

26. Wang, M. and Yao, J. (2021) Intertwined supply network design under facility and transportation disruption from the viability perspective, International Journal of Production Research, DOI: 10.1080/00207543.2021.1930237

Reviewer 3 Report

The paper presents an interesting research issue to establish the relation between farmers and service providers. The problem is well-formulated and a proper solution is proposed. However, more details about the genetic algorithm and its related formulas and parameters should be provided. 

Round 2

Reviewer 2 Report

The authors already responded my comments.

The presentation of this version has been improved.

However, the Abstract need to be improved. For example, 

"Finally, this study verifies the validity and feasibility of the model  and algorithm.". The case used in this study  should be described in the section of abstract.

I recommend it for publication with minor changes.
